# Validating International Classification of Disease 10<sup>th</sup> Revision algorithms for identifying influenza and respiratory syncytial virus hospitalizations

**Mackenzie A. Hamilton**[1,2], **Andrew Calzavara**[1], **Scott D. Emerson**[1], **Mohamed Djebli**[1,2], **Maria E. Sundaram**[1], **Adrienne K. Chan**[2,3,4], **Rafal Kustra**[2], **Stefan D. Baral**[5], **Sharmistha Mishra**[3,6,7,8], **Jeffrey C. Kwong**[1,2,9,10,11,12] *

1 ICES, Toronto, Ontario, Canada, 2 Dalla Lana School of Public Health, University of Toronto, Toronto, Ontario, Canada, 3 Institute of Health Policy, Management and Evaluation, University of Toronto, Toronto, Ontario, Canada, 4 Division of Infectious Diseases, Department of Medicine, Sunnybrook Health Sciences Centre, University of Toronto, Toronto, Ontario, Canada, 5 Department of Epidemiology, John Hopkins Bloomberg School of Public Health, Baltimore, Maryland, United States of America, 6 Department of Medicine, St. Michael's Hospital, University of Toronto, Toronto, Ontario, Canada, 7 Institute of Medical Sciences, University of Toronto, Toronto, Ontario, Canada, 8 MAP Centre for Urban Health Solutions, St. Michael's Hospital, Li Ka Shing Knowledge Institute, Toronto, Ontario, Canada, 9 Public Health Ontario, Toronto, Ontario, Canada, 10 Department of Family and Community Medicine, University of Toronto, Toronto, Ontario, Canada, 11 University Health Network, Toronto, Ontario, Canada, 12 Centre for Vaccine Preventable Diseases, University of Toronto, Toronto, Ontario, Canada

* jeff.kwong@utoronto.ca

## Abstract

### Objective

Routinely collected health administrative data can be used to efficiently assess disease burden in large populations, but it is important to evaluate the validity of these data. The objective of this study was to develop and validate International Classification of Disease 10<sup>th</sup> revision (ICD -10) algorithms that identify laboratory-confirmed influenza or laboratory-confirmed respiratory syncytial virus (RSV) hospitalizations using population-based health administrative data from Ontario, Canada.

### Study design and setting

Influenza and RSV laboratory data from the 2014–15, 2015–16, 2016–17 and 2017–18 respiratory virus seasons were obtained from the Ontario Laboratories Information System (OLIS) and were linked to hospital discharge abstract data to generate influenza and RSV reference cohorts. These reference cohorts were used to assess the sensitivity, specificity, positive predictive value (PPV) and negative predictive value (NPV) of the ICD-10 algorithms. To minimize misclassification in future studies, we prioritized specificity and PPV in selecting top-performing algorithms.

### Results

83,638 and 61,117 hospitalized patients were included in the influenza and RSV reference cohorts, respectively. The best influenza algorithm had a sensitivity of 73% (95% CI 72% to

**Data Availability Statement:** Data sharing agreements between ICES and the Ontario Ministry of Health and Long Term Care outlined in Ontario's

Personal Health Information Protection Act legally prohibit ICES from making the dataset publicly available. Therefore, due to our legally binding agreements, we cannot publicly share the dataset from this study. Qualified researchers can obtain access to the data required to replicate the analyses conducted in this study. One can request access to the data at https://www.ices.on.ca/DAS, by email at das@ices.on.ca, or by phone at 1-844-848-9855.

**Funding:** This study was funded by the Canadian Institutes of Health Research (JCK, PJT 159516, https://cihr-irsc.gc.ca/e/193.html; SM, VR5 172683; https://webapps.cihr-irsc.gc.ca/decisions/p/project_details.html?applId=430319&lang=en) and a St. Michael's Hospital Foundation Research Innovation Council's 2020 COVID-19 Research Award (SM; https://secure3.convio.net/smh/site/SPageNavigator/RIC2019.html). SM is supported by a Tier 2 Canada Research Chair in Mathematical Modelling and Program Science (CRC number 950-232643). JCK is supported by a Clinician-Scientist Award from the University of Toronto Department of Family and Community Medicine. This study was supported by ICES, which is funded by an annual grant from the Ontario Ministry of Health and Long-Term Care (MOHLTC). The funders had no role in study design, data collection and analysis, decision to publish, or preparation of the manuscript.

**Competing interests:** The authors have declared that no competing interests exist.

74%), specificity of 99% (95% CI 99% to 99%), PPV of 94% (95% CI 94% to 95%), and NPV of 94% (95% CI 94% to 95%). The best RSV algorithm had a sensitivity of 69% (95% CI 68% to 70%), specificity of 99% (95% CI 99% to 99%), PPV of 91% (95% CI 90% to 91%) and NPV of 97% (95% CI 97% to 97%).

## Conclusion

We identified two highly specific algorithms that best ascertain patients hospitalized with influenza or RSV. These algorithms may be applied to hospitalized patients if data on laboratory tests are not available, and will thereby improve the power of future epidemiologic studies of influenza, RSV, and potentially other severe acute respiratory infections.

---

## Introduction

Routinely collected health administrative data are increasingly being used to assess disease burden and aetiology [1,2]. Algorithms applied to International Classification of Disease (ICD) codes documented in hospital discharge abstracts can be used to identify cases of a disease for the purposes of disease surveillance, but it is imperative to evaluate the validity of such algorithms to limit misclassification bias in epidemiologic studies.

While several studies have assessed the validity of ICD codes for identifying influenza and respiratory syncytial virus (RSV) within health administrative data [1–8], many of those studies had limitations. Some studies could only examine correlative patterns between true cases and ICD-coded cases at an aggregate level, because they could not link data at the individual level [2,3,5,6]. Without individual-level data, there remains the risk of misclassification of individual cases, as well as challenges in characterizing the sensitivity, specificity, and predictive values of these algorithms. When individual-level data were available and validity parameters were reported, studies were generally limited by one or more of: small numbers of study centres, restricted participant age ranges, or inclusion of few respiratory virus seasons [1,4,7,8]. Consequently, the generalizability of these algorithms is uncertain.

The objective of this study was to develop and validate more generalizable ICD 10th revision (ICD-10) case-finding algorithms to identify patients hospitalized with laboratory-confirmed influenza or laboratory-confirmed RSV using population-based health administrative data from Ontario, Canada.

## Methods

### Ethical considerations

This study used laboratory and health administrative data from Ontario, Canada (population 13.5 million in 2016) housed at ICES. ICES is a prescribed entity under section 45 of Ontario's Personal Health Information Protection Act (PHIPA). Section 45 authorizes ICES to collect personal health information, without consent, for the purpose of analysis or compiling statistical information with respect to the management of, evaluation or monitoring of, the allocation of resources to or planning for all or part of the health system. Projects conducted under section 45, by definition, do not require review by a Research Ethics Board. This project was conducted under section 45, and was approved by ICES' Privacy and Legal Office.

## Data sources

Ontario's universal healthcare system captures virtually all healthcare interactions. To identify eligible patients for this study, we used data from the Ontario Laboratories Information System (OLIS), the Canadian Institute for Health Information's Discharge Abstract Database (CIHI-DAD), and the Registered Persons Database (RPDB). These datasets were linked using unique encoded identifiers and analyzed at ICES.

OLIS is an electronic repository of Ontario's laboratory test results, containing information on laboratory orders, patient demographics, provider information, and test results. The system captures data from hospital, commercial, and public health laboratories participating in OLIS. OLIS excludes: tests performed for purposes other than providing direct care to patients; tests that are ordered for out-of-province patients or providers; and tests for patients with health cards that are recorded as lost, stolen, expired, or invalid.

Implemented in 1988, CIHI-DAD captures administrative, clinical, and demographic information on all hospitalization discharges. Following a patient's discharge from hospital, a trained medical coder codes the medical record with up to 25 ICD-10 diagnosis codes (1 "most responsible" diagnosis code and up to 24 additional diagnosis codes), all of which are recorded in CIHI-DAD.

The RPDB provides basic demographic information on all individuals who have ever had provincial health insurance, including birth date, sex and postal code of residence. Ontario health insurance eligibility criteria are summarized in Table A of the S1 Appendix.

## Generating influenza and RSV reference standard cohorts

Influenza and RSV polymerase chain reaction (PCR) laboratory data were obtained from OLIS over 4 respiratory virus seasons ranging from 2014–15 to 2017–18. This time frame was selected to include as many seasons as possible during a period when a relatively higher and stable proportion of laboratories were reporting to OLIS. Respiratory virus seasonality was defined to create the most inclusive time frames that would capture influenza and RSV seasonal activity in Ontario between the 2014–15 and 2017–18 viral seasons according to data provided by Public Health Ontario's Respiratory Pathogen Bulletin [9]. Therefore, influenza tests were collected from November to May and RSV tests were collected from November to April. Only one test per person per season was included in the reference cohort. If an individual was tested multiple times per season, we included the first positive test, or the first negative test if all tests were negative. Tests were excluded if they were linked to an individual who: was missing information on birth date, sex, or postal code from the RPDB; was not eligible for provincial health insurance or resided out of province according to the RPDB; or had a death date registered before the specimen collection date.

Laboratory data were then linked to CIHI-DAD hospitalization data using patients' unique encoded identifiers. Only patients with suspected community-acquired infections, defined as specimen collection within 3 days before or after a hospital admission, were included in the analysis. This definition ensured reference hospitalizations were more likely to be associated with community-acquired influenza or RSV infection. Individuals with suspected nosocomial infections, defined as hospitalizations associated with specimens collected more than 72 hours post admission [10], were excluded from the reference cohorts for that respective season. Overall, the "true positive" influenza and RSV reference cohorts comprised all hospitalized patients who tested positive for influenza or RSV by PCR within 3 days of admission, respectively, and the "true negative" influenza and RSV reference cohorts comprised all hospitalized patients who tested negative for influenza or RSV by PCR within 3 days of admission, respectively.

## Statistical analysis

The reference cohorts were used to assess the validity of influenza and RSV case-finding algorithms. Algorithms were defined according to combinations of ICD-10 codes that have been previously described in the literature [1,4,5] (see Table B in S1 Appendix for the detailed list of ICD-10 codes). In brief, algorithms included virus-specific ICD-10 codes alone (influenza: J09, J10.0, J10.1, J10.8; RSV: J12.1, J20.5, J21.0, B97.4) or in combination with common acute respiratory infection outcome codes such as pneumonia (J12.8, J12.9), bronchitis (J20.8, J20.9), or bronchiolitis (J21.8, J21.9).

The validity of each algorithm was evaluated by calculating sensitivity, specificity, positive predictive value (PPV), and negative predictive value (NPV). First, validity parameters were calculated by evaluating the "most responsible diagnosis" code in the discharge abstract. If an ICD-10 code in the algorithm was recorded as the most responsible diagnosis in the discharge abstract, then it was classified as an algorithm-positive record. Next, validity parameters were calculated using all diagnosis codes available in the discharge abstract. If an ICD-10 code in the algorithm was recorded as any diagnosis code on the discharge abstract, then it was classified as an algorithm-positive record. Algorithms applied to the most responsible diagnosis code were consistently less accurate than the same algorithms applied to all diagnosis codes (see Tables A–D in S2 Appendix). Therefore, we present the results of the latter analyses only.

To minimize false positive rates and minimize misclassification of algorithm-positive cases, top-performing algorithms were selected according to specificity and PPV parameters [11]. If multiple algorithms had similar specificity and PPV, we then prioritized sensitivity. Since PPV and NPV are susceptible to changes in disease prevalence [12], and thus may vary depending on patient age or month of hospital admission, we also validated the top-performing algorithms in the reference cohorts stratified by age and month of hospital admission. The algorithms with consistently high specificity and PPV were selected as top-performing algorithms.

We calculated 95% confidence intervals using the Clopper-Pearson exact method [13]. All analyses were conducted using SAS version 9.4 (SAS Institute, Cary, NC, USA).

## Results

### Influenza and RSV reference cohorts

We identified 133,422 and 96,624 PCR testing events for influenza and RSV, respectively, in OLIS during the 2014–15 to 2017–18 respiratory virus seasons (Fig 1). After exclusions, 83,638 (63%) and 61,117 (63%) events for influenza and RSV, respectively, were associated with a hospitalization within 3 days of specimen collection and thus comprised the reference cohorts. Reference cohort characteristics are summarized in Table 1. True positive cases, defined as hospitalizations associated with a positive PCR test, comprised 17.6% of the influenza cohort and 9.2% of the RSV cohort (Table 1). Patient age ranged from 0 to 105 years. In both reference cohorts, all age strata had at least 2,000 patients.

### Algorithm validation

Most influenza and RSV ICD-10 algorithms had specificities ≥95% and NPVs ≥94%. Algorithm sensitivities and PPVs were more variable, ranging from 69% to 91% and 20% to 94% respectively (Tables 2 and 3). We established two highly accurate ICD-10 algorithms that identified influenza hospitalizations: one that found discharge abstracts with influenza-specific codes accompanied by laboratory confirmation of influenza (FLU1; ICD-10 Codes: J09, J10.0, J10.1, J10.8) and another that found discharge abstracts with influenza-specific codes with or

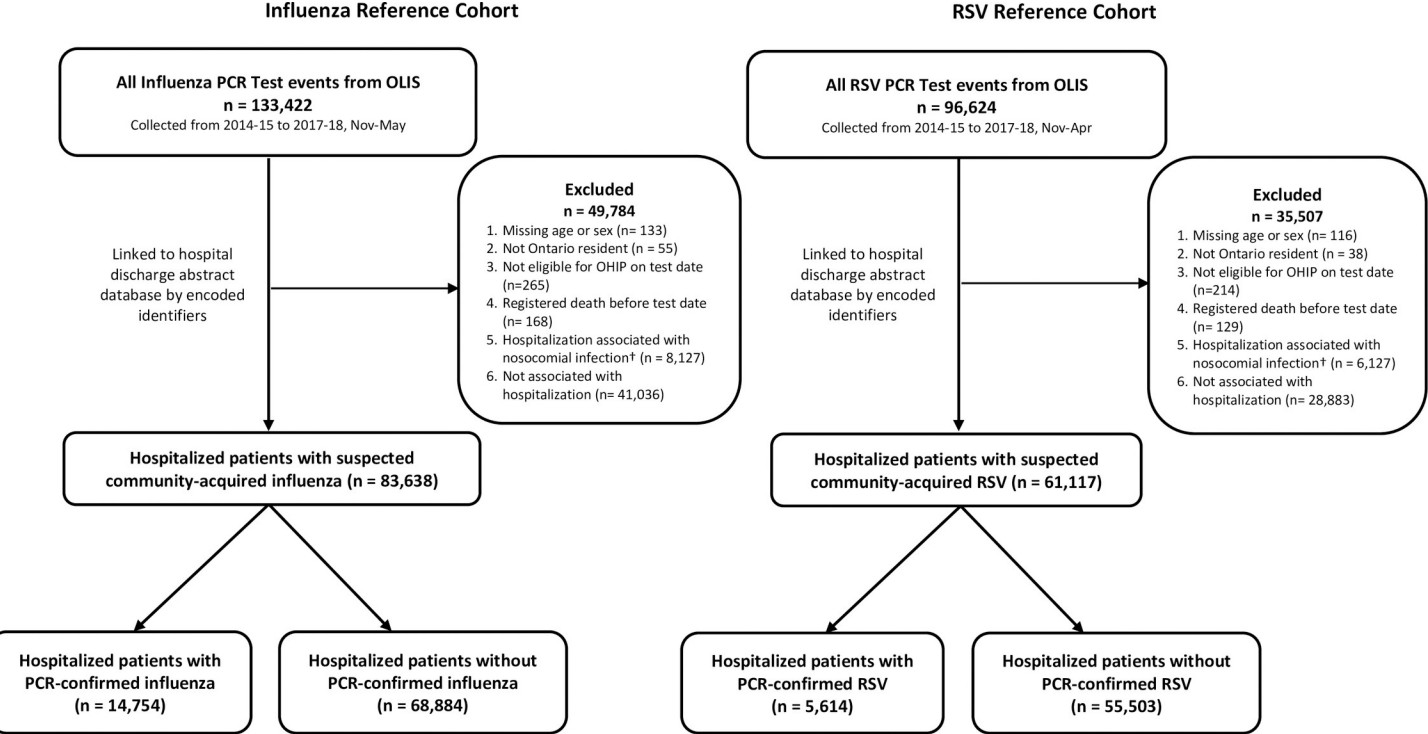

**Fig 1. Flow diagram of patients included and excluded in the influenza and RSV algorithm development cohorts.** PCR, polymerase chain reaction; OLIS, Ontario Laboratory Information System; RSV, respiratory syncytial virus; OHIP, Ontario Health Insurance Plan. †Nosocomial infections were defined as hospitalizations associated with specimen collection dates more than 3 days post hospital admission and before hospital discharge. Patients with nosocomial associated infections were only excluded for the season in which their first positive test event was defined as a nosocomial infection.

without laboratory confirmation of influenza (FLU2; ICD-10 Codes: J09, J10.0, J10.1, J10.8, J11.0, J11.1, J11.8). Specificity was ≥98% and PPV was ≥91% for both algorithms (Table 2).

Similarly, we established two highly accurate ICD-10 algorithms that identified RSV hospitalizations: one that found discharge abstracts with RSV-specific codes (RSV1; ICD-10 Codes: J12.1, J20.5, J21.0, B97.4), and another that found discharge abstracts with RSV-specific codes and unspecified acute lower respiratory tract infection codes (RSV2; ICD-10 Codes: J12.1, J20.5, J21.0, B97.4, J22). Specificity was 99%, and PPV was ≥87% for both algorithms (Table 3).

## Algorithm validation by age group and month of admission

Validity of the FLU1 and FLU2 algorithms did not vary substantially by age (Table 4). Both algorithms had specificities ≥98% and PPVs ≥89% across all age strata. More variability in FLU1 and FLU2 algorithm validity was observed when assessed by month of hospital admission (Table E in S2 Appendix). Specificity of both algorithms remained ≥98% during all months, whereas sensitivity and PPV decreased in November and May.

RSV1 and RSV2 algorithm validity was more variable across age strata (Table 4). Algorithm specificities were ≥94% across all age strata, while algorithm sensitivities were higher among children aged 0–4 years (e.g. RSV1 Sensitivity = 76%) compared to adults (e.g. adults aged 20–49 years, RSV1 Sensitivity = 49%). Further, PPVs declined among patients aged 5–19 years to lows of 85% for RSV1 and 78% for RSV2. RSV1 and RSV2 algorithm validity also varied by month of hospital admission (Table E in S2 Appendix). Algorithm specificities were ≥99% for November through April, while algorithm sensitivities and PPVs declined in April (RSV1: Sensitivity = 56% PPV = 89%; RSV2: Sensitivity = 57% PPV = 81%).

**Table 1. Characteristics of the influenza and RSV reference cohorts.**

| Characteristics | Influenza Reference Cohort (N = 83,638) | RSV Reference Cohort (N = 61,117) |
|---|---|---|
| **Virus detected by PCR, n (%)** | 14,754 (17.6%) | 5,614 (9.2%) |
| **Season, n (%)** | | |
| 2014–2015 | 14,344 (17.2%) | 9,267 (15.2%) |
| 2015–2016 | 14,931 (17.9%) | 9,435 (15.4%) |
| 2016–2017 | 23,081 (27.6%) | 20,360 (33.3%) |
| 2017–2018 | 31,282 (37.4%) | 22,055 (36.1%) |
| **Age Group, n (%)** | | |
| 0–4 | 10,173 (12.2%) | 7,260 (11.9%) |
| 5–19 | 2,890 (3.5%) | 2,058 (3.4%) |
| 20–34 | 3,185 (3.8%) | 2,400 (3.9%) |
| 35–49 | 4,890 (5.8%) | 3,661 (6.0%) |
| 50–64 | 12,572 (15.0%) | 9,297 (15.2%) |
| 65–74 | 14,332 (17.1%) | 10,695 (17.5%) |
| 75–84 | 17,879 (21.4%) | 12,999 (21.3%) |
| 85+ | 17,717 (21.2%) | 12,747 (20.9%) |
| **Sex on RPDB, n (%)** | | |
| Female | 41,997 (50.2%) | 30,655 (50.2%) |
| Male | 41,641 (49.8%) | 30,462 (49.8%) |
| **Neighborhood Income Quintile, n (%)** | | |
| Missing Data | 237 (0.3%) | 179 (0.3%) |
| 1 (lowest) | 22,238 (26.6%) | 16,145 (26.4%) |
| 2 | 18,726 (22.4%) | 13,998 (22.9%) |
| 3 | 16,190 (19.4%) | 12,107 (19.8%) |
| 4 | 13,433 (16.1%) | 9,580 (15.7%) |
| 5 (highest) | 12,814 (15.3%) | 9,108 (14.9%) |
| **Risk factors for serious viral infection, n (%)** | | |
| Asthma | 24,662 (29.5%) | 18,239 (29.8%) |
| Chronic Obstructive Pulmonary Disease | 23,812 (28.5%) | 17,183 (28.1%) |
| Immunodeficiency | 7,461 (8.9%) | 5,693 (9.3%) |
| Cancer | 9,920 (11.9%) | 7,527 (12.3%) |
| Diabetes | 29,384 (35.1%) | 21,710 (35.5%) |
| Hypertension | 52,656 (63.0%) | 38,696 (63.3%) |
| Cardiac Ischemic Disease | 17,065 (20.4%) | 12,560 (20.6%) |
| Congestive Heart Failure | 25,031 (29.9%) | 18,525 (30.3%) |
| Ischemic Stroke or Transient Ischemic Attack | 7,322 (8.8%) | 5,408 (8.8%) |
| Advanced Liver Disease | 2,958 (3.5%) | 2,313 (3.8%) |
| Chronic Kidney Disease | 19,316 (23.1%) | 14,458 (23.7%) |
| Dementia or frailty score > 15, n (%) | 21,508 (25.7%) | 16,084 (26.3%) |
| **LTC Home Resident, n (%)†** | 6,704 (8.0%) | 5,001 (8.2%) |
| **Received Influenza Vaccination, n (%)†** | 28,469 (34.0%) | 20,450 (33.5%) |
| **Prior Hospital Admissions, mean (SD) ‡** | 1.77 (2.64) | 1.84 (2.75) |
| **Prior Physician Visits, mean (SD) §** | 14.71 (13.40) | 15.09 (13.67) |
| **Length of Hospital Stay, days, mean (SD)** | 8.94 (18.24) | 9.22 (18.93) |

(*Continued*)

**Table 1.** (Continued)

| Characteristics | Influenza Reference Cohort (N = 83,638) | RSV Reference Cohort (N = 61,117) |
|---|---|---|
| Spent time in ICU, n (%) | 15,753 (18.8%) | 11,728 (19.2%) |

Continuous variables are expressed as means and standard deviations. Categorical variables are expressed as absolute numbers and percentages. One hospitalization per person, per season was included in counts. RSV, respiratory syncytial virus; PCR, polymerase chain reaction; RPDB, Registered Persons Database; LTC, long-term care; ICU, intensive care unit.

† As recorded in the same season as hospitalization.

‡ Mean prior hospital admissions in the past 3 years.

§ Mean prior physician visits in the past year.

**Table 2. Validation of ICD-10 algorithms to identify hospitalized individuals with influenza infection.**

| ICD-10 Algorithm | TP | FP | FN | TN | Sensitivity (95% CI) | Specificity (95% CI) | PPV (95% CI) | NPV (95% CI) |
|---|---|---|---|---|---|---|---|---|
| Influenza-specific codes[a], **FLU1**[*] | 10,755 | 653 | 3,999 | 68,231 | 0.73(0.72–0.74) | 0.99(0.99–0.99) | 0.94(0.94–0.95) | 0.94(0.94–0.95) |
| Influenza-specific + Influenza (virus not identified)[b], **FLU2**[*] | 12,245 | 1,201 | 2,509 | 67,683 | 0.83(0.82–0.84) | 0.98(0.98–0.98) | 0.91(0.91–0.92) | 0.96(0.96–0.97) |
| Influenza-specific + ARI of multiple/unspecified sites[c] | 10,965 | 3,337 | 3,789 | 65,547 | 0.74(0.74–0.75) | 0.95(0.95–0.95) | 0.77(0.76–0.77) | 0.95(0.94–0.95) |
| Influenza-specific + viral pneumonia[d] | 10,819 | 1,341 | 3,935 | 67,543 | 0.73(0.73–0.74) | 0.98(0.98–0.98) | 0.89(0.88–0.90) | 0.94(0.94–0.95) |
| Influenza-specific + bronchopneumonia[e] | 11,517 | 18,690 | 3,237 | 50,194 | 0.78(0.77–0.79) | 0.73(0.73–0.73) | 0.38(0.38–0.39) | 0.94(0.94–0.94) |
| Influenza-specific + acute bronchitis[f] | 10,804 | 1,194 | 3,950 | 67,690 | 0.73(0.73–0.74) | 0.98(0.98–0.98) | 0.90(0.90–0.91) | 0.94(0.94–0.95) |
| Influenza-specific + acute bronchiolitis[g] | 10,792 | 2,117 | 3,962 | 66,767 | 0.73(0.72–0.74) | 0.97(0.97–0.97) | 0.84(0.83–0.84) | 0.94(0.94–0.95) |
| Influenza-specific + ARI of multiple sites + acute bronchitis + acute bronchiolitis | 12,321 | 3,201 | 2,433 | 65,683 | 0.84(0.83–0.84) | 0.95(0.95–0.96) | 0.79(0.79–0.80) | 0.96(0.96–0.97) |
| Influenza-specific + viral infection (unspecified site)[h] | 10,904 | 2,518 | 3,850 | 66,366 | 0.74(0.73–0.75) | 0.96(0.96–0.96) | 0.81(0.81–0.82) | 0.95(0.94–0.95) |
| Influenza-specific + unspecified acute lower respiratory tract infection[i] | 10,792 | 1,033 | 3,962 | 67,851 | 0.73(0.72–0.74) | 0.99(0.98–0.99) | 0.91(0.91–0.92) | 0.94(0.94–0.95) |
| Influenza-specific + all general ARI codes[j] | 13,379 | 26,040 | 1,375 | 42,844 | 0.91(0.90–0.91) | 0.62(0.62–0.63) | 0.34(0.33–0.34) | 0.97(0.97–0.97) |

ICD-10, International Classification of Disease 10[th] Revision; ARI, acute respiratory infection; TP, true positive; FP, false positive; FN, false negative; TN, true negative; PPV, positive predictive value; NPV, negative predictive value

[*]Identified as a top-performing algorithm.

a—Influenza-specific (virus identified) ICD-10 codes: J09, J10.0, J10.1, J10.8

b—Influenza (virus not identified) ICD-10 codes: J11.0, J11.1, J11.8

c—Acute upper respiratory infections of multiple unspecified sites (virus unspecified/not identified) ICD-10 codes: J06.0, J06.8, J06.9

d—Viral pneumonia (virus unspecified/not identified) ICD-10 codes: J12.8, J12.9

e—Bronchopneumonia (organism unspecified) ICD-10 codes: J18.0, J18.8, J18.9

f—Acute bronchitis (organism unspecified) ICD-10 codes: J20.8, J20.9

g—Acute bronchiolitis (organism unspecified) ICD-10 codes: J21.8, J21.9

h–Viral infection (unspecified site) ICD-10 code: B34

i—Unspecified acute lower respiratory tract infection ICD-10 code: J22

j—General ARI ICD-10 codes: J11.0 J11.1, J11.8, J06.0, J06.8, J06.9, J12.8, J12.9, J18.0, J18.8, J18.9, J20.8, J20.9, J21.8, J21.9, B34, J22

**Table 3. Validation of ICD-10 algorithms to identify hospitalized individuals with RSV infection.**

| ICD-10 Algorithm | TP | FP | FN | TN | Sensitivity (95% CI) | Specificity (95% CI) | PPV (95% CI) | NPV (95% CI) |
|---|---|---|---|---|---|---|---|---|
| RSV-specific codes[a], **RSV1**[*] | 3,881 | 403 | 1,733 | 55,100 | 0.69(0.68–0.70) | 0.99(0.99–0.99) | 0.91(0.90–0.91) | 0.97(0.97–0.97) |
| RSV-specific + ARI of multiple/unspecified sites[b] | 4,038 | 2,079 | 1,576 | 53,424 | 0.72(0.71–0.73) | 0.96(0.96–0.96) | 0.66(0.65–0.67) | 0.97(0.97–0.97) |
| RSV-specific + Influenza (virus not identified)[c] | 3,895 | 1,551 | 1,719 | 53,952 | 0.69(0.68–0.71) | 0.97(0.97–0.97) | 0.72(0.70–0.73) | 0.97(0.97–0.97) |
| RSV-specific + viral pneumonia[d] | 3,938 | 867 | 1,676 | 54,636 | 0.70(0.69–0.71) | 0.98(0.98–0.99) | 0.82(0.81–0.83) | 0.97(0.97–0.97) |
| RSV-specific + bronchopneumonia[e] | 4,280 | 13,667 | 1,334 | 41,836 | 0.76(0.75–0.77) | 0.75(0.75–0.76) | 0.24(0.23–0.24) | 0.97(0.97–0.97) |
| RSV-specific + acute bronchitis[f] | 3,896 | 764 | 1,718 | 54,739 | 0.69(0.68–0.71) | 0.99(0.99–0.99) | 0.84(0.83–0.85) | 0.97(0.97–0.97) |
| RSV-specific + acute bronchiolitis[g] | 4,115 | 990 | 1,499 | 54,513 | 0.73(0.72–0.74) | 0.98(0.98–0.98) | 0.81(0.80–0.82) | 0.97(0.97–0.97) |
| RSV-specific + ARI of multiple sites + acute bronchitis + acute bronchiolitis | 4,276 | 3,007 | 1,338 | 52,496 | 0.76(0.75–0.77) | 0.95(0.94–0.95) | 0.59(0.58–0.60) | 0.98(0.97–0.98) |
| RSV-specific + viral infection (unspecified site)[h] | 3,944 | 1,650 | 1,670 | 53,853 | 0.70(0.69–0.71) | 0.97(0.97–0.97) | 0.71(0.69–0.72) | 0.97(0.97–0.97) |
| RSV-specific + unspecified acute lower respiratory tract infection[i], **RSV2**[*] | 3,896 | 598 | 1,718 | 54,905 | 0.69(0.68–0.71) | 0.99(0.99–0.99) | 0.87(0.86–0.88) | 0.97(0.97–0.97) |
| RSV-specific + all general ARI codes[j] | 4,769 | 18,867 | 845 | 36,636 | 0.85(0.84–0.86) | 0.66(0.66–0.66) | 0.20(0.20–0.21) | 0.98(0.98–0.98) |

ICD-10, International Classification of Disease 10[th] Revision; RSV, respiratory syncytial virus; ARI, acute respiratory infection; TP, true positive; FP, false positive; FN, false negative; TN, true negative; PPV, positive predictive value; NPV, negative predictive value.

[*]Identified as a top performing algorithm.

a—RSV-specific (virus identified) ICD-10 codes: J12.1, J20.5, J21.0, B97.4

b—Acute upper respiratory infections of multiple unspecified sites (virus unspecified/not identified) ICD-10 codes: J06.0, J06.8, J06.9

c—Influenza (virus not identified) ICD-10 codes: J11.0, J11.1, J11.8

d—Viral pneumonia (virus unspecified/not identified) ICD-10 codes: J12.8, J12.9

e—Bronchopneumonia (organism unspecified) ICD-10 codes: J18.0, J18.8, J18.9

f—Acute bronchitis (organism unspecified) ICD-10 codes: J20.8, J20.9

g—Acute bronchiolitis (organism unspecified) ICD-10 codes: J21.8, J21.9

h–Viral infection (unspecified site) ICD-10 code: B34

i—Unspecified acute lower respiratory tract infection ICD-10 code: J22

j—General ARI ICD-10 codes: J11.0 J11.1, J11.8, J06.0, J06.8, J06.9, J12.8, J12.9, J18.0, J18.8, J18.9, J20.8, J20.9, J21.8, J21.9, B34, J22

Overall, the FLU1 algorithm and the RSV1 algorithm maintained the highest specificity and PPV across all age strata and months of admission, and were therefore classified as the most valid algorithms to identify influenza and RSV hospitalizations.

## Discussion

We established two highly specific ICD-10 algorithms to identify influenza and RSV hospitalizations using large, population-based reference cohorts of patients with laboratory-confirmed hospitalizations over four respiratory virus seasons. Based on the criteria of specificity and PPV, the most valid influenza algorithm included all influenza-specific ICD-10 codes that included laboratory confirmation (FLU1), while the most valid RSV algorithm included all RSV-specific ICD-10 codes (RSV1).

**Table 4. Validation of top-performing ICD-10 influenza and RSV algorithms by age at hospital admission.**

| ICD-10 Algorithm | TP | FP | FN | TN | Sensitivity (95% CI) | Specificity (95% CI) | PPV (95% CI) | NPV (95% CI) |
|---|---|---|---|---|---|---|---|---|
| **FLU1 Algorithm[a]** | | | | | | | | |
| 0–4 | 751 | 69 | 378 | 8,975 | 0.67(0.64–0.69) | 0.99(0.99–0.99) | 0.92(0.89–0.93) | 0.96(0.96–0.96) |
| 5–19 | 379 | 24 | 223 | 2,264 | 0.63(0.59–0.67) | 0.99(0.98–0.99) | 0.94(0.91–0.96) | 0.91(0.90–0.92) |
| 20–34 | 295 | 15 | 156 | 2,719 | 0.65(0.61–0.70) | 0.99(0.99–1.00) | 0.95(0.92–0.97) | 0.95(0.94–0.95) |
| 35–49 | 518 | 28 | 228 | 4,116 | 0.69(0.66–0.73) | 0.99(0.99–1.00) | 0.95(0.93–0.97) | 0.95(0.94–0.95) |
| 50–64 | 1,405 | 78 | 544 | 10,545 | 0.72(0.70–0.74) | 0.99(0.99–0.99) | 0.95(0.93–0.96) | 0.95(0.95–0.95) |
| 65–74 | 1,772 | 115 | 652 | 11,793 | 0.73(0.71–0.75) | 0.99(0.99–0.99) | 0.94(0.93–0.95) | 0.95(0.94–0.95) |
| 75–84 | 2,683 | 160 | 931 | 14,105 | 0.74(0.73–0.76) | 0.99(0.99–0.99) | 0.94(0.93–0.95) | 0.94(0.93–0.94) |
| 85+ | 2,952 | 164 | 887 | 13,714 | 0.77(0.76–0.78) | 0.99(0.99–0.99) | 0.95(0.94–0.95) | 0.94(0.94–0.94) |
| **FLU2 Algorithm[b]** | | | | | | | | |
| 0–4 | 844 | 104 | 285 | 8,940 | 0.75(0.72–0.77) | 0.99(0.99–0.99) | 0.89(0.87–0.91) | 0.97(0.97–0.97) |
| 5–19 | 438 | 39 | 164 | 2,249 | 0.73(0.69–0.76) | 0.98(0.98–0.99) | 0.92(0.89–0.94) | 0.93(0.92–0.94) |
| 20–34 | 347 | 38 | 104 | 2,696 | 0.77(0.73–0.81) | 0.99(0.98–0.99) | 0.90(0.87–0.93) | 0.96(0.96–0.97) |
| 35–49 | 598 | 64 | 148 | 4,080 | 0.80(0.77–0.83) | 0.98(0.98–0.99) | 0.90(0.88–0.92) | 0.97(0.96–0.97) |
| 50–64 | 1,595 | 173 | 354 | 10,450 | 0.82(0.80–0.84) | 0.98(0.98–0.99) | 0.90(0.89–0.92) | 0.97(0.96–0.97) |
| 65–74 | 2,003 | 213 | 421 | 11,695 | 0.83(0.81–0.84) | 0.98(0.98–0.98) | 0.90(0.89–0.92) | 0.97(0.96–0.97) |
| 75–84 | 3,078 | 275 | 536 | 13,990 | 0.85(0.84–0.86) | 0.98(0.98–0.98) | 0.92(0.91–0.93) | 0.96(0.96–0.97) |
| 85+ | 3,342 | 295 | 497 | 13,583 | 0.87(0.86–0.88) | 0.98(0.98–0.98) | 0.92(0.91–0.93) | 0.96(0.96–0.97) |
| **RSV1 Algorithm[c]** | | | | | | | | |
| 0–4 | 2,072 | 241 | 639 | 4,308 | 0.76(0.75–0.78) | 0.95(0.94–0.95) | 0.90(0.88–0.91) | 0.87(0.86–0.88) |
| 5–19 | 71 | 13 | 71 | 1,903 | 0.50(0.42–0.59) | 0.99(0.99–1.00) | 0.85(0.75–0.91) | 0.96(0.95–0.97) |
| 20–49 | 98 | 13 | 100 | 5,850 | 0.49(0.42–0.57) | 1.00(1.00–1.00) | 0.88(0.81–0.94) | 0.98(0.98–0.99) |
| 50–64 | 257 | 25 | 190 | 8,825 | 0.57(0.53–0.62) | 1.00(1.00–1.00) | 0.91(0.87–0.94) | 0.98(0.98–0.98) |
| 65–74 | 353 | 31 | 233 | 10,078 | 0.60(0.56–0.64) | 1.00(1.00–1.00) | 0.92(0.89–0.94) | 0.98(0.97–0.98) |
| 75–84 | 470 | 52 | 261 | 12,216 | 0.64(0.61–0.68) | 1.00(0.99–1.00) | 0.90(0.87–0.92) | 0.98(0.98–0.98) |
| 85+ | 560 | 28 | 239 | 11,920 | 0.70(0.67–0.73) | 1.00(1.00–1.00) | 0.95(0.93–0.97) | 0.98(0.98–0.98) |
| **RSV2 Algorithm[d]** | | | | | | | | |
| 0–4 | 2,079 | 271 | 632 | 4,278 | 0.77(0.75–0.78) | 0.94(0.93–0.95) | 0.88(0.87–0.90) | 0.87(0.86–0.88) |
| 5–19 | 72 | 20 | 70 | 1,896 | 0.51(0.42–0.59) | 0.99(0.98–0.99) | 0.78(0.68–0.86) | 0.96(0.96–0.97) |
| 20–49 | 100 | 24 | 98 | 5,839 | 0.51(0.43–0.58) | 1.00(0.99–1.00) | 0.81(0.73–0.87) | 0.98(0.98–0.99) |
| 50–64 | 257 | 52 | 190 | 8,798 | 0.57(0.53–0.62) | 0.99(0.99–1.00) | 0.83(0.79–0.87) | 0.98(0.98–0.98) |
| 65–74 | 353 | 69 | 233 | 10,040 | 0.60(0.56–0.64) | 0.99(0.99–0.99) | 0.84(0.80–0.87) | 0.98(0.97–0.98) |
| 75–84 | 472 | 93 | 259 | 12,175 | 0.65(0.61–0.68) | 0.99(0.99–0.99) | 0.84(0.80–0.87) | 0.98(0.98–0.98) |
| 85+ | 563 | 69 | 236 | 11,879 | 0.70(0.67–0.74) | 0.99(0.99–1.00) | 0.89(0.86–0.91) | 0.98(0.98–0.98) |

ICD-10, International Classification of Disease 10th Revision; RSV, respiratory syncytial virus; TP, true positive; FP, false positive; FN, false negative; TN, true negative; PPV, positive predictive value; NPV, negative predictive value.

a—Influenza-specific ICD-10 codes with virus identified: J09, J10.0, J10.1, J10.8

b—Influenza-specific ICD-10 codes with and without virus identified: J09, J10.0, J10.1, J10.8, J11.0, J11.1, J11.8

c—RSV-specific ICD-10 codes with virus identified: J12.1, J20.5, J21.0, B97.4

d—RSV-specific ICD-10 codes with virus identified + unspecified acute lower respiratory tract infection ICD-10 code: J12.1, J20.5, J21.0, B97.4, J22

This finding was expected given our reference cohorts were defined using laboratory test results. Medical coding is performed at discharge when testing results may be available; thus, medical coding and laboratory data are not necessarily independent.

FLU1 and RSV1 maintained high specificity and PPV when the reference cohorts were stratified by age. Thus, the algorithms can be applied to paediatric, adult, and elderly populations with low risk of misclassification bias. The specificity of the algorithms also remained

high when assessed by month of hospitalization, although PPV was more variable. The PPV of FLU1 dropped to lows of 87% in November and 86% in May, while the PPV of RSV1 dropped to a low of 89% in April. These decreases were expected, as PPV is dependent on disease prevalence, and the decreases were concordant with typical declines in respiratory virus prevalence and activity in Ontario during those months [14]. Notably, the absolute number of false positives generated during times of low viral activity made up <8% of overall FLU1 false positives and <7% of overall RSV1 false positives. Therefore, while PPV declined during months of lower viral activity, the overall algorithm validity was not impacted.

Our findings concur with previous literature indicating that ICD-10 codes have high specificity and moderate sensitivity for identifying influenza and RSV hospitalizations using health administrative data [1,4,7,8]. Where direct comparisons are possible, our quantitative measures of specificity align with previous findings, while our measures of sensitivity are lower. For example, Moore et al. found that an algorithm that included codes for influenza with or without laboratory confirmation (J10.0-J10.9, J11.0-J11.9) had a specificity of 98.6% and a sensitivity of 86.1% for children aged 0–9 years, whereas our FLU2 algorithm had a specificity of 99% and a sensitivity of 73–75% for children aged 0–19 years [4]. Furthermore, Pisesky et al. found that an algorithm comprising RSV-specific codes (J12.1, J20.5, J21.0, B97.4) had a specificity of 99.6% and a sensitivity of 97.9% for children aged 0–3 years, whereas our RSV1 algorithm had corresponding values of 95% and 76% for children aged 0–4 years [1].

Distinctions between our study populations may explain the differences in sensitivity observed. Pisesky et al. studied a population from a specialized hospital in Ottawa, Ontario [1], while Moore et al. studied a Western Australian population [4]. In contrast, our study was conducted in a larger cohort of patients using data from hospitals across the entire province of Ontario. ICD-10 codes may be used more or less frequently across jurisdictions and institutions resulting in variable algorithm sensitivity. The discrepancies highlight the importance of validating algorithms within distinct populations.

While we established two highly specific algorithms that identify influenza and RSV hospitalizations, some limitations must be considered. First, our reference cohorts only included hospitalized patients who were tested by PCR for the respective pathogens and did not include patients who were not tested. Untested patients with suspected respiratory infections may differ from the tested population of patients. They may have less severe symptoms at hospitalization, may be more likely to live in long-term care facilities where outbreaks have occurred, or may be more likely to live in less-resourced settings where testing is limited. Untested patients may have had more pressing medical concerns at hospitalization and therefore testing was not a priority, or they may have been hospitalized at an overcrowded, high-volume site. Testing may further depend on the age of the patient at admission or the protocols in place at the hospital. By including hospitals across the entire province of Ontario we aimed to mitigate hospital-specific variability that may have affected the generalizability of our results. However, variability between untested and tested patients must be considered when using the algorithms to assess certain risk factors that may be associated with propensity to receive a test. For example, risk factors such as age, symptom severity, comorbidities, and residence in long-term care facilities may be associated with propensity to receive a PCR test, and thus may have biased estimates of effect when using these algorithms. Caution must also be applied when assessing risk factors in specific settings that have differing testing practises compared to general Ontario hospitals.

Another limitation is that our top-performing algorithms were selected to maximize specificity and PPV. This approach was taken to minimize misclassification of cases rather than non-cases. Depending on future study objectives, it may be more important to maximize sensitivity. For example, our algorithms significantly underestimate the number of true influenza

and RSV cases in the Ontario population, and thus would not be suitable to estimate population burden of influenza or RSV. Therefore, validity parameters have been reported for all algorithms tested to facilitate the selection of the best algorithm(s) for particular studies.

Use of these algorithms in non-Ontario-based cohorts also warrants caution. PPV and NPV are highly susceptible to changes in disease prevalence [12]. Coding practises and testing practises may vary across jurisdictions, affecting all validity measures reported [15,16]. Thus, it may be necessary to re-validate these algorithms when applying them to other populations.

Our findings have important implications for future studies that aim to assess the aetiology of severe outcomes for influenza and RSV hospitalizations using broad health administrative data. Not all hospitals across Ontario currently submit laboratory data to OLIS. Further, OLIS data collection was limited between 2007 and 2012 as laboratories only gradually started submitting data upon implementation of OLIS in 2007. As CIHI-DAD is available for all hospitals across Ontario, these algorithms will allow us to create larger and more representative cohorts of patients hospitalized with influenza or RSV, increasing the power of future aetiological studies. Lastly, since historical CIHI-DAD data are available as early as 1988, these algorithms could be used to assess changes in disease prevalence and aetiology over time.

## Conclusion

Using a population-based cohort of patients tested for influenza and RSV, we identified two highly specific algorithms that best ascertain paediatric, adult, and elderly patients hospitalized with influenza or RSV. These algorithms will improve future efforts to evaluate prognostic and aetiologic factors associated with influenza and RSV when reporting of laboratory data is limited. The same principles may be applicable for other severe acute respiratory infections.

## Supporting information

**S1 Appendix. Supplementary information for study methodology.**
(DOCX)

**S2 Appendix. Supplementary results.**
(DOCX)

## Acknowledgments

Parts of this material are based on data and information compiled and provided by the Ontario Ministry of Health and Long-Term Care (MOHLTC) and the Canadian Institute for Health Information (CIHI). The analyses, conclusions, opinions and statements expressed herein are solely those of the authors; no official endorsement by MOHLTC, or CIHI should be inferred.

## Author Contributions

**Conceptualization:** Mackenzie A. Hamilton, Jeffrey C. Kwong.

**Data curation:** Andrew Calzavara, Scott D. Emerson.

**Formal analysis:** Mackenzie A. Hamilton.

**Funding acquisition:** Sharmistha Mishra, Jeffrey C. Kwong.

**Methodology:** Mackenzie A. Hamilton, Andrew Calzavara, Scott D. Emerson, Mohamed Djebli, Maria E. Sundaram, Jeffrey C. Kwong.

**Supervision:** Adrienne K. Chan, Rafal Kustra, Stefan D. Baral, Sharmistha Mishra, Jeffrey C. Kwong.

**Writing – original draft:** Mackenzie A. Hamilton.

**Writing – review & editing:** Mackenzie A. Hamilton, Mohamed Djebli, Maria E. Sundaram, Adrienne K. Chan, Rafal Kustra, Stefan D. Baral, Sharmistha Mishra, Jeffrey C. Kwong.

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
