## [Decision Letter · Decision Letter 0]

11 Sep 2020

PONE-D-20-24822

Validating International Classification of Disease 10th Revision algorithms for identifying influenza and respiratory syncytial virus hospitalizations

PLOS ONE

Dear Dr. Hamilton,

Thank you for submitting your manuscript to PLOS ONE. After careful consideration, we feel that it has merit but does not fully meet PLOS ONE’s publication criteria as it currently stands. Therefore, we invite you to submit a revised version of the manuscript that addresses the points raised during the review process.

We look forward to receiving your revised manuscript.

Kind regards,

Judith Katzenellenbogen, Ph D

Academic Editor

PLOS ONE

Journal Requirements:

"This study was funded by the Canadian Institutes of Health Research (JCK; PJT

159516; https://cihr-irsc.gc.ca/e/193.html) and a St. Michael’s Hospital Foundation

Research Innovation Council’s 2020 COVID-19 Research Award (SM;

https://secure3.convio.net/smh/site/SPageNavigator/RIC2019.html). The funders had

no role in study design, data collection and analysis, decision to publish, or preparation

of the manuscript."

"SM is supported by a Tier 2 Canada Research Chair in Mathematical Modeling and Program Science. JCK is supported by a Clinician-Scientist Award from the University of Toronto Department of Family and Community Medicine. This study was supported by ICES, which is funded by an annual grant from the Ontario Ministry of Health and Long-Term Care (MOHLTC)."

Reviewers' comments:

Reviewer's Responses to Questions

**Comments to the Author**

1. Is the manuscript technically sound, and do the data support the conclusions?

Reviewer #1: Yes

Reviewer #2: Yes

Reviewer #3: Yes

2. Has the statistical analysis been performed appropriately and rigorously? 

Reviewer #1: Yes

Reviewer #2: Yes

Reviewer #3: Yes

3. Have the authors made all data underlying the findings in their manuscript fully available?

Reviewer #1: No

Reviewer #2: Yes

Reviewer #3: No

4. Is the manuscript presented in an intelligible fashion and written in standard English?

Reviewer #1: Yes

Reviewer #2: Yes

Reviewer #3: Yes

5. Review Comments to the Author

Reviewer #1: This is an important study addressing limitations in the existing literature about the validity of ICD-10 coded administrative data for influenza and RSV case ascertainment.

I have a few comments for the authors to address:

1. The algorithms do not take into account likely differences between hospitals. In a study with similar objectives for a different disease (reference below for your information), we found that inter-hospital differences were important in predicting the validity of case ascertainment through ICD-10 codes, because of implicit differences in de facto coding practices between hospitals and over time (i.e. ultimately medical coders). While there are coding guidelines, there is likely still some variability in how individual coders apply these rules in practice. At a minimum this hypothesis should be evaluated as part of algorithm development. In our study, we used hospital random effects to capture facility-specific differences, but other approaches are possible.

2. You state that the algorithms using all diagnosis positions were more accurate. Could you comment on the differences in accuracy between algorithms using all diagnosis positions and algorithms using "most responsible diagnosis" only? I think, the finding that algorithms should use all diagnosis positions is an interesting finding in itself and should be supported with the evidence that it is based on.

3. A large number of persons were excluded from the cohorts due to hospital acquired disease. I agree with the importance to exclude records for these episodes of disease to identify community acquired disease, but I am unsure of the need to exclude the persons' records entirely. Were all records for a person who ever had hospital acquired influenza excluded and if so, why did you make this choice? Intuitively, the records for these persons should be used, at a minimum for other seasons, especially since this cohort might be systematically different (possibly more comorbidities, older, overall more susceptible etc.) from people who never had hospital-acquired influenza.

Regarding data availability, the authors have specified why their data is not fully publicly available.

References:

Bond-Smith D, Seth R, de Klerk N, Nedkoff L, Anderson M, Hung J, Cannon J, Griffiths K, Katzenellenbogen JM. Development and Evaluation of a Prediction Model for Ascertaining Rheumatic Heart Disease Status in Administrative Data. Clin Epidemiol. 2020;12:717-730

https://doi.org/10.2147/CLEP.S241588

Reviewer #2: The aim of the study was to identify valid algorithms for identifying influenza and RSV associated infections resulting in hospitalisation. This is a noble and important goal, since often times, population-based researchers do not have access to laboratory data. The algorithms developed as part of this study will be useful for future research. I have just a few points where I though additional consideration or clarification would be helpful:

1. Because the reference cohort was selected based on laboratory data, by nature, untested individuals are excluded. This is a major limitation of the study, since those who are tested vs. not can be very different. For example, tested individuals could have presented with much more severe infection or may represent certain population groups of interest. I appreciate the authors mention this (in brief) on page 15, but I feel this point requires more attention. It is not just residents of long-term care facilities who may be less likely to be tested, testing varies by a number of other factors. I think this should be discussed in more detail - particularly with regard to how this may influence the study results and generalizability.

2. Line 288: I didn’t quite follow the statement: “Differences in reference cohort definitions directly influence the resulting algorithm validity” – while this is certainly true, I didn’t understand how the authors linked this statement to the justification for inclusion of PCR-only results. Based on a positive immunofluorescence or culture results, we could still confidently say that an individual had influenza. While it makes sense to exclude serology, the absence of non-PCR testing for virus detection could introduce some selection bias. Is this what the authors meant? Please confirm (Also note that Western Australia (data published by Moore et al.) mostly performs testing by PCR, similar to Ontario.)

3. Could the authors provide more detail on the definition of influenza and RSV seasons? These were selected independently, which is appropriate, but I wasn’t sure whether RSV season typically extended months after influenza season – or if in some winters, RSV activity started before influenza season. Some further information here would be helpful, since as the authors point out – seasonality can strongly influence the validity of diagnostic coding.

4. It makes sense to me that the best performing algorithm are the codes where virus is identified, as the laboratory data and the coding are not necessarily independent. Medical coding is performed at discharge, when testing results may already be available. I think this is worth mentioning in the interpretation of findings.

Reviewer #3: This manuscript evaluates the validity of ICD10 codes associated with influenza- and RSV-related hospital admissions and has identified algorithms (of ICD-10 codes) that best identify patients associated with these two infections. Although there have been several publications about the use of ICD codes to identify specific respiratory infection-related hospitalisation across the world, this paper uses the best data source and provides evidence for the Ontario region. I don’t have any concerns with regards to the data used or the analysis. However, as the authors have right pointed out in the manuscript, due to differences in coding (and even laboratory testing) practices between hospitals/jurisdictions/regions/countries, the generalisability of the identified algorithms to non-Ontario based cohorts is limited.

I just have one very minor comment: The authors have looked at four respiratory virus seasons 2014-2015, 2015-12016, 2016-2017 and 207-2018. However, reading the abstract and the methods/results gives the impression that only two seasons (2014-2015 and 2017-2018) were used or compared. The authors might want to make this clear.

6. PLOS authors have the option to publish the peer review history of their article (what does this mean?). If published, this will include your full peer review and any attached files.

Reviewer #1: **Yes: **Daniela Bond-Smith

Reviewer #2: No

Reviewer #3: No

---

## [Author Response · Author response to Decision Letter 0]

5 Nov 2020

Dear Reviewers and Editors,

We thank you for your time, comments and expertise. Please find a point-by-point response to your comments below. 

Academic Editor: 

Comment 1: Please ensure that your manuscript meets PLOS ONE’s style requirements, including those for file naming. 

Response 1: We have updated the manuscript to follow PLOS ONE’s style requirements. Notably, we have re-formatted the headings, we have re-formatted the figure and table titles, we have updated the file names and we have added a supporting information section at the end of the manuscript.

Comment 2: We note that you have indicated that data from this study are available upon request. PLOS only allows data to be available upon request if there are legal or ethical restrictions on sharing data publicly. For information on unacceptable data access restrictions, please see http://journals.plos.org/plosone/s/data-availability#loc-unacceptable-data-access-restrictions.

b) If there are no restrictions, please upload the minimal anonymized data set necessary to replicate your study findings as either Supporting Information files or to a stable, public repository and provide us with the relevant URLs, DOIs, or accession numbers. Please see http://www.bmj.com/content/340/bmj.c181.longfor guidelines on how to de-identify and prepare clinical data for publication. For a list of acceptable repositories, please see http://journals.plos.org/plosone/s/data-availability#loc-recommended-repositories. 

Response 2: The dataset from this study is held securely in coded form at ICES. Data sharing agreements between ICES and the Ontario Ministry of Health and Long Term Care outlined in Ontario’s Personal Health Information Protection Act legally prohibit ICES from making the dataset publicly available, as it may contain personally identifiable information. Therefore, due to our legally binding agreements, we cannot publicly share the dataset from this study. Certain individuals may be granted access to the data if they meet pre-specified criteria for confidential access. One can request access to the data from this study at www.ices.on.ca/DAS. One can also contact ICES Data & Analytic Services by email at das@ices.on.ca, or by phone at 1-844-848-9855. This information has been outlined in our cover letter.

Comment 3: We note that you have provided funding information that is not currently declared in your Funding Statement. However, funding information should not appear in the Acknowledgments section or other areas of your manuscript. We will only publish funding information present in the Funding Statement section of the online submission form. Please remove any funding-related text from the manuscript and let us know how you would like to update your Funding Statement.

Response 3: Thank you for highlighting this discrepancy. We have outlined an updated funding statement in our cover letter. One piece of funding related text remains in our acknowledgements statement. Our organization, ICES, is an independent, non-profit research institute funded by an annual grant from the Ontario Ministry of Health and Long-Term Care (MOHLTC). Through our agreements with the MOHLTC, we are required to mention the MOHLTC’s funding support for ICES in our acknowledgments. We do not feel that it is appropriate to place this text in our funding statement because the MOHLTC did not directly fund this study.

Reviewer #1: This is an important study addressing limitations in the existing literature about the validity of ICD-10 coded administrative data for influenza and RSV case ascertainment.

Comment 1: The algorithms do not take into account likely differences between hospitals. In a study with similar objectives for a different disease (reference below for your information), we found that inter-hospital differences were important in predicting the validity of case ascertainment through ICD-10 codes, because of implicit differences in de facto coding practices between hospitals and over time (i.e. ultimately medical coders). While there are coding guidelines, there is likely still some variability in how individual coders apply these rules in practice. At a minimum this hypothesis should be evaluated as part of algorithm development. In our study, we used hospital random effects to capture facility-specific differences, but other approaches are possible.

Response 1: We thank the reviewer for this important comment. Inter-hospital differences across Ontario must be considered for individual-level predictive analyses. As the reviewer described, it is plausible that variability in coding and testing practises may influence the algorithm validity at a single center scale. Importantly, our algorithms are not meant to be applied to single centers, but rather a province-wide hospital discharge abstract database to generate broad cohorts of Ontario patients hospitalized with influenza or respiratory syncytial virus. We have identified the most valid algorithms to be applied across the entire province. We acknowledge that the most valid algorithms for single centers may differ. This is an interesting research question to examine, but it extends beyond the current research objectives. We highlight this limitation in the discussion of our manuscript, stating it may be necessary to re-validate the algorithms when applying them to populations that differ from that in our study. 

Comment 2: You state that the algorithms using all diagnosis positions were more accurate. Could you comment on the differences in accuracy between algorithms using all diagnosis positions and algorithms using "most responsible diagnosis" only? I think, the finding that algorithms should use all diagnosis positions is an interesting finding in itself and should be supported with the evidence that it is based on.

Response 2: We thank the reviewer for highlighting the lack of clarity in our statement. Algorithms using the most responsible diagnosis code had marginally better specificity and positive predictive values (PPV), and substantially worse sensitivity, as compared to algorithms that used all medical diagnosis codes. The claim that “algorithms using all diagnosis codes were more accurate than algorithms using the most responsible diagnosis code only” was based on calculations of Youden’s index and Cohen’s kappa. We found that the Youden’s index was always larger when applying algorithms to all diagnosis codes compared to the most responsible diagnosis code. We further found that all algorithms (with the exception of our most general influenza algorithm and most general RSV algorithm) had larger Cohen’s kappa values when applying them to all diagnosis codes compared to the most responsible diagnosis code. We have provided algorithm validity when applied to the most responsible diagnosis code in Tables A and B in the S2 Appendix. A comparison of Youden’s index and Cohen’s kappa for algorithms applied to the most responsible diagnosis code versus all diagnosis codes can be seen in Table C and D of the S2 appendix. The claim in the manuscript has also been modified to improve transparency. 

Comment 3: A large number of persons were excluded from the cohorts due to hospital acquired disease. I agree with the importance to exclude records for these episodes of disease to identify community acquired disease, but I am unsure of the need to exclude the persons' records entirely. Were all records for a person who ever had hospital acquired influenza excluded and if so, why did you make this choice? Intuitively, the records for these persons should be used, at a minimum for other seasons, especially since this cohort might be systematically different (possibly more comorbidities, older, overall more susceptible etc.) from people who never had hospital-acquired influenza.

Response 3: We thank the reviewer for highlighting this important point. In our study, individuals with hospital acquired disease were only excluded during the respiratory season in which they were identified as having hospital-acquired disease. Therefore, if these individuals were hospitalized with community-acquired disease in another season, their hospitalization event was included for that other season. We have provided additional clarity in our manuscript, and in the legend of Figure 1.

Individuals with hospital-acquired influenza or RSV were excluded for the entire respective season of infection because we did not have a way to differentiate secondary or tertiary tests associated with the initial nosocomial infection from secondary or tertiary tests that could have resulted from re-infection via interactions within their community. 

Reviewer #2: The aim of the study was to identify valid algorithms for identifying influenza and RSV associated infections resulting in hospitalisation. This is a noble and important goal, since often times, population-based researchers do not have access to laboratory data. The algorithms developed as part of this study will be useful for future research. I have just a few points where I thought additional consideration or clarification would be helpful:

Comment 1: Because the reference cohort was selected based on laboratory data, by nature, untested individuals are excluded. This is a major limitation of the study, since those who are tested vs. not can be very different. For example, tested individuals could have presented with much more severe infection or may represent certain population groups of interest. I appreciate the authors mention this (in brief) on page 15, but I feel this point requires more attention. It is not just residents of long-term care facilities who may be less likely to be tested, testing varies by a number of other factors. I think this should be discussed in more detail - particularly with regard to how this may influence the study results and generalizability.

Response 1: We thank the reviewer for their positive comments and feedback. In the discussion of this study’s limitations, we have provided more detail on how the selection of our reference cohort may affect the generalizability of the study results and bias future effect estimates.

Comment 2: Line 288: I didn’t quite follow the statement: “Differences in reference cohort definitions directly influence the resulting algorithm validity” – while this is certainly true, I didn’t understand how the authors linked this statement to the justification for inclusion of PCR-only results. Based on a positive immunofluorescence or culture results, we could still confidently say that an individual had influenza. While it makes sense to exclude serology, the absence of non-PCR testing for virus detection could introduce some selection bias. Is this what the authors meant? Please confirm (Also note that Western Australia (data published by Moore et al.) mostly performs testing by PCR, similar to Ontario.)

Response 2: We thank the reviewer for highlighting the lack of clarity. The statement in line 288 was not meant to highlight selection bias by excluding non-PCR testing. Instead, it was meant to summarize that our reference cohorts varied from previous studies based on: the age of the patients; the population from which the patients were identified; and the testing methods used to define true positive patients from true negative patients. Differences in all validity parameters may arise due to variability in any of these factors. For example, younger patients may be more likely to receive a test regardless of the severity of their symptoms – this may increase the proportion of individuals in the true negative cohort, most likely influencing the negative predictive value. Testing sensitivity and specificity depends on when a specimen is collected, how the specimen is collected, the type of laboratory test used and the specific location where the test is run. At a specialized center, patients may be more likely to receive a test earlier rather than later, improving the probability that an individual who truly has an infection falls in the “true positive” cohort – this would most likely improve algorithm positive predictive values. Finally, ICD-10 codes may be used more or less frequently across jurisdictions and at particular institutions influencing the sensitivity and specificity of an ICD-10 algorithm in differing populations.

We realize that some of this discussion is not relevant to our direct analysis of the differing sensitivities observed. Further, it is unlikely that test sensitivity and specificity were drastically different between the specialized hospital in Ottawa, Western Australian hospitals, and Ontario hospitals. Therefore, we have edited this section to strictly discuss differing sensitivity observed.

Comment 3: Could the authors provide more detail on the definition of influenza and RSV seasons? These were selected independently, which is appropriate, but I wasn’t sure whether RSV season typically extended months after influenza season – or if in some winters, RSV activity started before influenza season. Some further information here would be helpful, since as the authors point out – seasonality can strongly influence the validity of diagnostic coding.

Response 3: Respiratory virus seasonality was defined according to Public Health Ontario’s Respiratory Pathogen Bulletin. We created the most inclusive time frames that would capture seasonal influenza and RSV activity in Ontario between the 2014-15 and 2017-18 seasons according to information accessed on Public Heath Ontario’s Respiratory Pathogen Bulletin. We have updated this section of the manuscript to define seasonality more clearly.

Comment 4: It makes sense to me that the best performing algorithms are the codes where virus is identified, as the laboratory data and the coding are not necessarily independent. Medical coding is performed at discharge, when testing results may already be available. I think this is worth mentioning in the interpretation of findings.

Response 4: Thank you for your comment. We agree that our findings were expected and have included a paragraph mentioning this in the interpretation of our findings. 

Reviewer #3: This manuscript evaluates the validity of ICD10 codes associated with influenza- and RSV-related hospital admissions and has identified algorithms (of ICD-10 codes) that best identify patients associated with these two infections. Although there have been several publications about the use of ICD codes to identify specific respiratory infection-related hospitalisation across the world, this paper uses the best data source and provides evidence for the Ontario region. I don’t have any concerns with regards to the data used or the analysis. However, as the authors have right pointed out in the manuscript, due to differences in coding (and even laboratory testing) practices between hospitals/jurisdictions/regions/countries, the generalisability of the identified algorithms to non-Ontario based cohorts is limited.

Comment 1: The authors have looked at four respiratory virus seasons 2014-2015, 2015-2016, 2016-2017 and 2017-2018. However, reading the abstract and the methods/results gives the impression that only two seasons (2014-2015 and 2017-2018) were used or compared. The authors might want to make this clear.

Response 1: We thank the reviewer for their positive comments. We have updated the abstract to clarify which seasons were included in our analyses.

We look forward to your response.

Yours truly,

Jeff Kwong, on behalf of the authors

---

## [Decision Letter · Decision Letter 1]

16 Dec 2020

Validating International Classification of Disease 10th Revision algorithms for identifying influenza and respiratory syncytial virus hospitalizations

PONE-D-20-24822R1

Dear Dr. Hamilton,

We’re pleased to inform you that your manuscript has been judged scientifically suitable for publication and will be formally accepted for publication once it meets all outstanding technical requirements.

Kind regards,

Judith Katzenellenbogen, Ph D

Academic Editor

PLOS ONE

Additional Editor Comments (optional):

Reviewers' comments:

Reviewer's Responses to Questions

**Comments to the Author**

1. If the authors have adequately addressed your comments raised in a previous round of review and you feel that this manuscript is now acceptable for publication, you may indicate that here to bypass the “Comments to the Author” section, enter your conflict of interest statement in the “Confidential to Editor” section, and submit your "Accept" recommendation.

Reviewer #1: All comments have been addressed

Reviewer #3: All comments have been addressed

2. Is the manuscript technically sound, and do the data support the conclusions?

Reviewer #1: Yes

Reviewer #3: (No Response)

3. Has the statistical analysis been performed appropriately and rigorously? 

Reviewer #1: Yes

Reviewer #3: (No Response)

4. Have the authors made all data underlying the findings in their manuscript fully available?

Reviewer #1: Yes

Reviewer #3: (No Response)

5. Is the manuscript presented in an intelligible fashion and written in standard English?

Reviewer #1: Yes

Reviewer #3: (No Response)

6. Review Comments to the Author

Reviewer #1: My comments have been adequately addressed. The paper lays out a methodological approach for developing a prediction algorithm, with its findings being highly specific to the data that it was developed for. It provides a useful contribution for others who may wish to develop a similar algorithm, but the its findings may not generalize beyond the original data.

Reviewer #3: (No Response)

7. PLOS authors have the option to publish the peer review history of their article (what does this mean?). If published, this will include your full peer review and any attached files.

Reviewer #1: No

Reviewer #3: No

---

## [Editor Report · Acceptance letter]

28 Dec 2020

PONE-D-20-24822R1 

Validating International Classification of Disease 10^th^ Revision algorithms for identifying influenza and respiratory syncytial virus hospitalizations 

Dear Dr. Hamilton:

I'm pleased to inform you that your manuscript has been deemed suitable for publication in PLOS ONE. Congratulations! Your manuscript is now with our production department. 

Kind regards, 

on behalf of

Dr. Judith Katzenellenbogen 

Academic Editor

PLOS ONE